# Examining Health Literacy in Taiwanese Smoking Cessation Populations: A Multidimensional Evaluation of Sociodemographic Factors and Domain-Specific Competencies

**DOI:** 10.3390/healthcare11162350

**Published:** 2023-08-21

**Authors:** Chi-Wei Lin, Wei-Hsuan Lin, Wei-Chieh Hung, Yi-Che Lee, Yi-Ching Yang, Ying-Wei Wang, Ching-Jung Ho, Tao-Qian Tang

**Affiliations:** 1Department of Family and Community Medicine, E-Da Hospital, I-Shou University, Kaohsiung City 84001, Taiwan; ed104283@edah.org.tw; 2School of Medicine for International Students, College of Medicine, I-Shou University, Kaohsiung City 84001, Taiwan; ed107698@edah.org.tw (W.-C.H.); ed103910@edah.org.tw (Y.-C.L.); 3Department of Family and Community Medicine, E-Da Hospital, Kaohsiung City 84001, Taiwan; angel.w.h.lin@gmail.com; 4Department of Family Medicine, E-Da Hospital, Kaohsiung City 84001, Taiwan; 5Division of Nephrology, Department of Internal Medicine, E-Da Hospital, Kaohsiung City 84001, Taiwan; 6Department of Family Medicine, National Cheng Kung University Hospital, Tainan City 704302, Taiwan; yiching@mail.ncku.edu.tw; 7Department of Family Medicine, College of Medicine, National Cheng Kung University, Tainan City 70401, Taiwan; 8Department of Family Medicine, Hualien Tzu Chi Hospital, Hualien City 970473, Taiwan; ywwang@mail.tcu.edu.tw; 9Department of Medical Humanities, School of Medicine, Tzu Chi University, Hualien City 97004, Taiwan; 10Graduate Institute of Adult Education, National Kaohsiung Normal University, Kaohsiung City 80201, Taiwan; 11School of Medicine, College of Medicine, I-Shou University, Kaohsiung City 84001, Taiwan; 12International Intercollegiate Ph.D. Program, National Tsing Hua University, Hsinchu City 30013, Taiwan

**Keywords:** health literacy, smoking cessation, sociodemographic factors

## Abstract

Background: Cigarette smoking is a serious global health issue. Limited studies previously analyzed health literacy components in patients undergoing smoking cessation interventions. This study focuses on individuals enrolled in smoking cessation services and investigates the distribution of health literacy in three domains (health care, disease prevention, and health promotion) and four abilities (access, understand, appraise, and apply health information). The study also explores the correlation between background factors (age, BMI, etc.) and health literacy, as well as the differences in health literacy levels among different background variables (gender, etc.). Methods: 228 individuals completed the health literacy questionnaire. Descriptive statistical analysis, Pearson Correlation, and a Chi-Squared Test were employed to investigate the various health literacy levels and background variables. Results: 68% had excellent or sufficient health literacy. A total of 32% were considered problematic or to have inadequate health literacy. Of the three domains of health literacy, participants performed better in the healthcare domain. More than one-third were problematic in accessing and appraising information. Conclusions: this paper, being the pilot study in providing an analysis of health literacy components in individuals undergoing smoking cessation, could serve as a useful reference for devising interventions for different population groups in trying to maximize successful cessation rates.

## 1. Introduction

According to The Tobacco Atlas: Fifth Edition, over 100 million people worldwide died directly from smoking in the 20th century [1]. The Health Promotion Administration in Taiwan surveyed adult smoking behavior in 2020. This survey showed 13.1% of adults over the age of 18 were smokers, the smoking rate was higher among men (23.1%) than among women (2.9%), the smoking rate among men peaked in the 46–50 age group (39.7%), and the smoking rate among women peaked in the 21–25 age group (7.5%) [2]. Fortunately, smokers in Taiwan have started to seek smoking cessation interventions in recent years, for which the main reason for quitting smoking is health factors (53.8%), such as fear of getting sick and negative effects on the fetus, followed by family and peer factors (14.7%) and smoker’s economic factors (7.7%). However, entering a medical institution to complete the smoking cessation process is never easy. An individual needs to possess the ability to obtain relevant information, master the correct knowledge of smoking cessation, evaluate the impact of various smoking cessation aids or activities by himself, and determine whether he can start to seek smoking cessation until he finally achieves smoking cessation successfully. These all involve core competencies and knowledge in various health literacy fields, affecting people’s intention to seek smoking cessation services and their chances of successfully quitting after receiving such services [2].

Health literacy implies the achievement of a level of knowledge, personal skills, and confidence to take action to improve individual and community health by changing unique lifestyles and living conditions. Thus, health literacy means more than being able to read pamphlets and make appointments. Enhancing people’s access to health-related information and their competence in utilizing it efficiently is paramount for empowerment and a crucial aspect of health literacy [3]. Several significant organizations have attempted to define health literacy. The American Medical Association (AMA) defines health literacy as a set of skills, including performing fundamental reading and numerical tasks necessary to navigate the healthcare environment [4]. On the other hand, the European Union (EU) defines health literacy as the capacity to access and use health-related information effectively to promote and maintain good health [5]. Similarly, the U.S. Department of Health and Human Services (HHS) defines health literacy as how individuals can obtain, process, and comprehend basic health information required to make informed health decisions [6].

Health literacy encompasses engaging in daily self-care and effectively managing chronic diseases. Gazmararian et al. surveyed 653 chronic disease patients over 65 regarding their health literacy and knowledge of chronic disease. They found that patients with inadequate health literacy were markedly inferior in the understanding of their diseases than those with adequate health literacy [7]. Robat Sarpooshi et al. surveyed 400 diabetes mellitus patients regarding their health literacy and self-care behaviors and found that patients with sufficient health literacy had better daily self-care behaviors [8].

Different personal sociodemographic factors and societal cultures may also affect individual health literacy. A survey in the United States in 2003 found that women had slightly better health literacy than men, Caucasian and Asian Americans had higher health literacy than did Hispanic, Indian and African Americans, people over the age of 65 had markedly poorer health literacy, and individuals with lower educational attainment had poorer health literacy [9]. Hoover et al. conducted a longitudinal cohort study on 1467 African Americans, and results showed that low health literacy was significantly associated with smoking, poorer psychological status, and higher perceived stress [10]. A study examining the health literacy, health behaviors, and self-health care of elderly adults showed that poorer health literacy was associated with a higher smoking rate [11]. However, such an association was not found in other research [12]. Certain investigations have examined the relationship between health literacy and a variety of health behaviors and outcomes, inclusive of smoking habits. Li et al. posited that the propensity for continued smoking amongst middle-aged Japanese individuals with a history of smoking is significantly associated with their lower health literacy, as evaluated using the Japanese Communicative and Critical Health Literacy Scale [13]. Likewise, Panahi et al. found a correlation between lower health literacy levels and decreased likelihood of adopting preventative smoking measures, especially notable among female students, students whose fathers lack formal education, and those already engaged in smoking habits [14]. One notable research conducted in a tertiary hospital in Turkey sought to assess the impact of patients’ health literacy levels on the effectiveness of smoking cessation treatment. Utilizing the European Health Literacy Survey Questionnaire (HLS-EU-Q) survey instrument, it was determined that 67.1% of the patient population exhibited inadequate or problematic general health literacy [15]. Nonetheless, as of the current literature, a definitive consensus remains elusive with respect to the link between low health literacy and health risk behaviors. Moreover, a scarcity of empirical research has successfully established health literacy as a standalone predictor for the initiation of smoking habits or suboptimal cessation outcomes [16].

This study focuses on individuals enrolled in smoking cessation services. It investigates the distribution of health literacy in three domains (health care, disease prevention, and health promotion) and four abilities (access, understand, appraise, and apply health information) among this population. The study also explores the correlation between background factors (age, BMI, smoking years, exhaled carbon monoxide (CO) level, Fagerström Test for Nicotine Dependence (FTND) score and health literacy, as well as the differences in health literacy levels among different background variables (gender, education level, marital status, obesity, alcohol consumption, betel nut use). The findings can be used as a reference for designing future smoking cessation educational materials and services tailored to different smoking populations with different backgrounds.

## 2. Materials and Methods

### 2.1. Study Population

The present research encompassed individuals aged 20 years and above who were beneficiaries of outpatient, inpatient, or workplace smoking cessation services at a major teaching hospital in southern Taiwan. The study design excluded those with any progressive ailment that might culminate in a life expectancy of less than half a year, as well as individuals exhibiting cognitive, visual, or auditory impairments where an available caregiver could not help. This exclusion criterion was determined by the potential impact these disabilities might have on the effective implementation of healthcare services. The participant pool was further limited to those who expressed a willingness to participate in interviews and formalized their agreement by signing consent forms.

The data collection period for this investigation spanned from January 2017 to December 2017. A comprehensive pool of 243 individuals who satisfied the inclusion criteria was solicited for participation; however, only 228 individuals acquiesced to participate and successfully completed the health literacy questionnaire on the day of their enrollment in the study.

### 2.2. Sociodemographics

The sociodemographic variables collected in this study included age, gender, educational attainment (primary school diploma or lower, junior high school diploma, senior high (vocational) school diploma, junior college degree, bachelor’s degree, graduate degree or above), and marital status (married, single, other).

### 2.3. Health Literacy Assessment

To assess health literacy, this study employed the Mandarin version of the European Health Literacy Survey Questionnaire (HLS-EU-Q) developed by Sørensen et al. [17]. The reliability and validity of this version of the HLS-EU-Q have been demonstrated previously through a survey of 2989 Taiwanese people. In terms of validity, the consistency of various external indicators with the theoretical framework was demonstrated through confirmatory factor analysis [18]. The questionnaire has 47 questions, covering three domains, i.e., health care, disease prevention, and health promotion. Each domain is classified into four competencies: access/obtain, understand, appraise/judge/evaluate, and apply/use health information (Appendix A).

The HLS-EU-Q uses a self-rating 4-point Likert scale, and the total score is the sum of the scores for all health literacy items. The higher the total score represents the better the respondent’s health literacy. To transform the HLS-EU-Q total score and three domain scores into a 0–50 point range, the following formula was applied: Index = (Mean − 1) × (50/3), where “Index” is the specific calculated index, “Mean” represents the average of all items for each person, “1” is the lowest possible mean value, “3” is the mean range, and “50” is the selected maximum value for the new metric. In the final index, 0 signifies the lowest health literacy (HL) level, while 50 denotes the highest [19].

The participants’ health literacy levels were categorized into four tiers based on their HLS-EU-Q scores. A total or subscale score ranging from 0 to 25 indicated inadequate health literacy, signifying that respondents perceived more than half of the items as extremely challenging. A score exceeding 25 but not surpassing 33 corresponded to problematic health literacy; within the 33–42 range was representative of sufficient health literacy, while scores above 42 were classified as the highest level of health literacy, denoted as excellent [19].

### 2.4. Analysis

After collecting the questionnaires, coding and archiving were carried out immediately. The descriptive statistical analysis examined the research subjects’ background information and health literacy distribution. Pearson Correlation was utilized to investigate the relationship between the health literacy score and sociodemographic variables (age, BMI, smoking frequency, CO levels, and FTND total score). Pearson’s Chi-Squared Test or Fisher’s Exact Test was employed to analyze the differences in health literacy levels among participants with various sociodemographic variables (gender, education, marital status, obesity, alcohol consumption, and betel nut chewing). The statistical analyses were performed using SPSS Statistics Version 22 (IBM Corp, released 2013; IBM SPSS Statistics for Windows, version 22.0; Armonk, NY, USA, IBM Corp).

## 3. Results

### 3.1. Sociodemographic Data of the Participants

This study had 228 individuals participating in a smoking cessation program, consisting of 209 males and 19 females. This male-to-female ratio was reflective of the gender composition of smokers within the broader Taiwanese population, thereby ensuring the representativeness of the sample. The mean age of the participants was 47.1 years, with approximately 75.9% reporting as married. Educational attainment for the majority of participants was Junior high school diploma or above. Within the examined sample, obesity, as defined by body mass index (BMI), was identified in 67 participants. Further, 35 individuals reported weekly alcohol consumption, while betel nut chewing habits were reported by 21 participants. The mean level of exhaled carbon monoxide (CO) among the participants was observed to be 18.0 parts per million (ppm). Furthermore, the mean score on the FTND was calculated to be 5.2 points.

In the evaluation conducted within this study, the participants’ overall health literacy demonstrated a mean score of 36.0 ± 6.2. Upon examining the three primary domains of health literacy, the healthcare domain displayed the highest score (37.8 ± 7.3), while the disease prevention domain yielded the lowest score (33.9 ± 7.6). With respect to the four discrete health literacy competencies, the aptitude for understanding information was the highest-ranked (37.7 ± 7.2), whereas the capacity to access information registered the lowest score (34.9 ± 6.7) (Table 1).

### 3.2. Distribution of the Health Literacy Levels of the Participants

In this study, more than half of the participants (50.9%) had sufficient health literacy, 17.1% had excellent health literacy, 28.5% were problematic, and only 3.5% had inadequate health literacy. In terms of the three domains of health literacy, the participants performed better in the healthcare domain, with approximately 75% being sufficient and excellent combined and 25% being inadequate and problematic. However, in the disease prevention domain, the performance was barely satisfactory, with just over half (53.1%) being sufficient and excellent and nearly half of the participants being inadequate and problematic. Additionally, among the four health literacy skills, there were more people with sufficient ability to apply information. In contrast, over one-third of the participants were problematic in accessing and appraising information (Table 2).

### 3.3. Factors Related to the Health Literacy of the Participants

This study analyzed the degree of correlation between the HLS-EU-Q total score and the sociodemographic variables of the participants (age, BMI, years of smoking, CO, and FTND total score). The findings indicated that the HLS-EU-Q total score and the sociodemographic variables did not exhibit a strong correlation within this study (Table 3).

Moreover, this investigation assessed the disparities in health literacy levels among individuals with diverse backgrounds, encompassing factors such as gender, education, marital status, obesity, alcohol consumption, and betel nut usage. As illustrated in Table 4, a notable difference in health literacy levels was observed between married and unmarried individuals at the time of the study (*p* < 0.05). Nevertheless, no significant differences in health literacy levels were detected concerning gender, education, alcohol consumption, betel nut chewing, and obesity within this population.

## 4. Discussion

In a health literacy survey involving 7795 participants from eight European countries, the World Health Organization (WHO) European Regional Office discovered that individuals with inadequate health literacy constituted approximately 11–12% of the overall population in Europe. Furthermore, the combined proportion of people with problematic health literacy and those with inadequate health literacy accounted for approximately 50% of the overall population [20]. In a separate large-scale health literacy study conducted in Taiwan, Duong et al. reported that the mean score for general health literacy (GHL) was 34.4 [18], which is marginally lower than the average score obtained from the population in the present study (36.0 ± 6.2). Nevertheless, both scores signify sufficient health literacy levels.

A body of literature has provided evidence to suggest a significant positive association between low levels of health literacy and an elevated propensity for smoking in comparison to their never-smoking counterparts [10,14,21]. This relationship indicates that individuals with lower health literacy may be more susceptible to initiating and maintaining smoking behaviors. Additionally, similar studies have underscored a lower likelihood of successful smoking cessation among individuals possessing low health literacy [22], hinting at potential obstacles to quitting due to possible difficulties in comprehending and applying health-related information. Contrarily, a contrasting stream of research has reported that the relationships between health literacy and various smoking behaviors, including initiation, maintenance, and cessation, may not be statistically significant [23,24]. These studies suggest that factors other than health literacy might have a greater influence on these behaviors or that the relationship may be more complex and not adequately captured by simple measures of correlation or association.

In this investigation, the health literacy of the smoking cessation population was predominantly characterized as sufficient, with a mere 3.5% of participants demonstrating inadequate health literacy. It is postulated that most of the study’s population comprised individuals who actively pursued smoking cessation services at medical institutions, thereby possessing adequate health literacy and exemplifying a proactive approach to self-management of their health. Moreover, of the three major domains of health literacy evaluated in this study, the smoking cessation population exhibited the strongest performance in the healthcare domain. This may be ascribed to the fact that numerous participants initially sought medical treatment for acute or chronic conditions and subsequently enrolled in smoking cessation services, resulting in a certain degree of familiarity with healthcare and medical institutions. With respect to the four health literacy skills assessed in this study, the smoking cessation population displayed superior performance in comprehending and applying information. This could be attributed to their familiarity with interaction patterns involving healthcare professionals and the inherent nature of the decision to quit smoking, which involves translating beliefs into action, culminating in higher scores within these domains.

In previous research examining health literacy and background factors, Bostock and Steptoe discovered that low health literacy among middle-aged and older adults in the United Kingdom correlated with unhealthy lifestyle habits, such as smoking and alcohol consumption [25]. In a Turkish study encompassing 207 patients with diabetes, individuals with higher education levels, regular exercise routines, and elevated health literacy scores exhibited improved self-care management [26]. Within Asian populations, a survey of 1348 residents in remote Japanese areas revealed that those with higher health literacy were less inclined to smoke, engaged in more frequent exercise, and achieved adequate sleep duration. Nevertheless, no significant differences were observed in maintaining an average body mass index, moderate alcohol consumption, daily breakfast consumption, or abstaining from snacks between meals compared to individuals with lower health literacy [27]. Another study conducted in a northern Taiwanese hospital, involving 403 patients with a mean age of 44.90 ± 15.80 years, identified a positive correlation between health literacy and female gender, as well as a higher average monthly income. Patients demonstrated superior healthcare health literacy to the general population but exhibited poorer health literacy in disease prevention and health promotion [28].

In analyzing factors related to health literacy among the smoking cessation population in this study, the associations between health literacy levels and background variables were generally weak. The sole exception was the marital status variable, wherein married individuals displayed significantly higher health literacy than unmarried, divorced, or widowed individuals. According to some research [25], age may influence health literacy, but in our study population, even though married individuals are 14.6 years older than unmarried individuals, only married individuals showed statistically significant higher Health Literacy scores; age did not. The findings of this study suggest that marital status influences health literacy among the smoking cessation population. This may be ascribed to individuals in marital relationships frequently receiving reminders and support from their spouses concerning healthcare and health maintenance, which cultivates a sense of security and support. As a result, participants demonstrate more positive responses when completing self-assessment health literacy questionnaires. This observation aligns with prior research conducted on selected populations in the United States [29], older adults in China [30], and adults in Poland [31], all of which indicate a significant correlation between health literacy and marital status.

Stewart conducted a study using the Short Test of Functional Health Literacy in Adults (S-TOFHLA) to investigate the relationship between social support, health literacy, and depressive symptoms among 200 smokers who were low-socioeconomic status (SES), with an average age of 46.1 years, the racial composition of the participants included Non-Latino White, Black, and other. The study’s findings highlighted a tendency for participants with lower levels of health literacy to identify as Black, be unemployed, and have reduced income and education levels. Moreover, lower health literacy was inversely related to perceived social support and positively associated with elevated manifestations of depressive symptoms [32]. Conversely, research has indicated that low health literacy is correlated with feelings of loneliness, limited social interactions, and infrequent engagement in social activities. Yet, no significant association was identified between low health literacy and levels of social support, or between low health literacy and living arrangements (namely, living alone versus cohabitating with others) [33].

Previous studies have indicated that individuals with limited health literacy commonly express feelings of shame or guilt associated with their difficulties in understanding health information [34,35]. Thus, those with lower HL may experience isolation and perceive less available support. Therefore, individuals with lower health literacy may find advantages in seeking assistance and encouragement from their family members or friends in their treatment process.

In individuals actively seeking smoking cessation services at healthcare facilities, there is an inherent motivation for adequate disease prevention. Additionally, they must be familiar with obtaining various services from these facilities to be willing to accept such preventive healthcare services. Consequently, the distribution of health literacy among this population and the correlation between health literacy and background factors and other health risk factors may differ slightly from the results of health literacy surveys targeting the general community population. This is not unexpected. A more pronounced incidence of restricted health literacy was observed in the community-dwelling population as compared to individuals frequenting primary care facilities or hospitals. Literature from a systematic review suggests that interactions with healthcare professionals could potentially enhance patients’ health literacy skills [36]. Such interactions expose patients to frequently employed healthcare terminologies and allow healthcare professionals to assist patients in developing a more profound understanding of health-related information [36]. Lee conducted a study using the Behavioral Risk Factor Surveillance System Questionnaire to measure the health literacy of 614 adults from Minnesota [37]. The study’s results indicated a tendency for individuals with elevated health literacy to exhibit greater comfort in utilizing the internet for health-related information retrieval, demonstrate enhanced proficiency in discerning appropriate search parameters and strategies, and exhibit increased ease in interpreting the acquired information [37]. The study found that individuals with higher health literacy are more comfortable seeking out health-related information on the internet, are more adept at knowing what to search for and how to find it, and are more comfortable interpreting the information that they access [37].

While our study discovered that individuals actively seeking smoking cessation services tend to have higher health literacy levels compared to the general population, we identified the unmarried group as a high-risk segment with lower health literacy within this cohort. Determining how to provide appropriate advice and services that cater to the needs of this demographic from the onset could potentially enhance the overall effectiveness and satisfaction of the smoking cessation services. In this current study, while we do not have the results as to whether the different health literacy levels amongst the patients seeking smoking cessation have on the eventual smoking cessation success rates, we identified the difference of health literacy levels enrolled in smoking cessation programs, as this will help in designing smoking cessation programs catered to the different subgroups of these patients.

Our study possesses several limitations worth noting. Firstly, this investigation is largely predicated upon descriptive statistics and analysis in examining patients undergoing smoking cessation, warranting further research into the influence of these health literacy components on smoking cessation outcomes. Additionally, our sample size was skewed towards males (91.7% male (209) vs. 8.3% female (19)), reflecting the higher prevalence of smoking among males in Asian cultures. The Health Promotion Administration’s 2020 survey in Taiwan also corroborated this gender discrepancy in smoking rates. Our study also limited itself to a single medical center in Southern Taiwan rather than adopting a multicenter approach. Furthermore, our focus was predominantly on a population actively engaging in smoking cessation, potentially overlooking those not seeking healthcare or refusing smoking cessation, who likely exhibit lower health literacy. The underrepresentation of these groups in our study might limit the generalizability of our findings.

## 5. Conclusions

This study found that participants seeking smoking cessation services generally had sufficient health literacy, performing best in the healthcare domain and applying information. However, issues were noted in the disease prevention domain and with accessing and appraising information. Marital status affected health literacy levels, with single or widowed individuals potentially requiring additional support due to high healthcare needs. Therefore, tailored follow-up programs might improve health literacy and care outcomes, particularly for those without family support. This study provides insights for future interventions and policymaking, emphasizing the need for resource allocation consideration.

## Figures and Tables

**Table 1 healthcare-11-02350-t001:** Sociodemographic data of the participants.

Variable	Total Sample (N = 228)
Gender, *n* (%)	
Male	209 (91.7)
Female	19 (8.3)
Married ^1^, *n* (%)	
No	55 (24.1)
Yes	173 (75.9)
Educational attainment, *n* (%)	
Primary school diploma or lower	25 (11.0)
Junior high school diploma or above	203 (89.0)
Drinking ^2^, *n* (%)	
No	193 (84.6)
Yes	35 (15.4)
Chewing betel nut, *n* (%)	
No	207 (90.8)
Yes	21 (9.2)
Obesity ^3^, *n* (%)	
No	161 (70.6)
Yes	67 (29.4)
Age (years) ± SD	47.1 ± 11.8
BMI ^4^, Mean (SD)	25.3 (4.6)
CO value, Mean (SD)	18.0 (12.3)
FTND ^5^, Mean (SD)	5.2 (2.5)
HLS-EU-Q total health, Mean (SD)	36.0 (6.2)
Domain, Mean (SD)	
Health care	37.8 (7.3)
Disease prevention	33.9 (7.6)
Health promotion	36.0 (7.9)
Competence, Mean (SD)	
Access information	34.9 (6.7)
Understand information	37.7 (7.2)
Appraise information	35.3 (6.9)
Apply information	37.4 (6.9)

^1^ Married is defined as having a formal marriage relationship, and other refers to single, divorced, or widowed. ^2^ Drinking is defined as the regular consumption of alcoholic beverages more than once per week, not purely social drinking. ^3^ Obesity is defined as BMI ≥ 27. ^4^ BMI = body mass index. ^5^ FTND = Fagerström Test for Nicotine Dependence.

**Table 2 healthcare-11-02350-t002:** The distribution of the health literacy levels of the participants. (N = 228).

	Inadequate	Problematic	Sufficient	Excellent
HLS-EU-Q total score ^1^	8 (3.5)	65 (28.5)	116 (50.9)	39 (17.1)
Domain, *n* (%)				
Health care	6 (2.6)	51 (22.4)	102 (44.7)	69 (30.3)
Disease prevention	25 (11.0)	82 (36.0)	79 (34.7)	42 (18.4)
Health promotion	21 (9.2)	50 (21.9)	105 (46.1)	52 (22.8)
Competence, *n* (%)				
Access information	17 (7.5)	66 (29.0)	107 (46.9)	38 (16.7)
Understand information	9 (4.0)	51 (22.4)	96 (42.1)	72 (31.6)
Appraise information	13 (5.7)	70 (30.7)	105 (46.1)	40 (17.5)
Apply information	8 (3.5)	47 (20.6)	115 (50.4)	58 (25.4)

^1^ The health literacy of the participants was classified into 4 levels based on the HLS-EU-Q classification: ‘inadequate’ (0–25), ‘problematic’ (>25–33), ‘sufficient’ (>33–42) and ‘excellent’ (>42–50) health literacy.

**Table 3 healthcare-11-02350-t003:** Correlations between the HLS-EU-Q total score and sociodemographic variables.

	HLS-EU-Q Total Score	Age	BMI	Years of Smoking	CO Value	FTND Total Score
HLS-EU-Q total score	1					
Age	0.065	1				
BMI	0.050	−0.012	1			
Years of smoking	0.000	0.827 **	−0.014	1		
CO value	−0.077	−0.198 **	−0.029	−0.156 *	1	
FTND total score	−0.064	0.061	0.068	0.159 *	0.217 **	1

* *p* < 0.05; ** *p* < 0.01.

**Table 4 healthcare-11-02350-t004:** Differences in health literacy levels between participants with different values for sociodemographic variables. (N = 228).

Variable	Inadequate (*n* = 8)	Problematic (*n* = 65)	Sufficient (*n* = 116)	Excellent (*n* = 39)	*p*-Value
Gender (*n*)					
Male	8	58	109	34	0.412 ^5^
Female	0	7	7	5	
Married ^1^ (*n*)					
No	2	24	22	7	0.040 *^,4^
Yes	6	41	94	32	
Educational attainment (*n*)					
Primary school diploma or lower	1	9	11	4	0.747 ^5^
Junior high school diploma or above	7	56	105	35	
Drinking ^2^ (*n*)					
No	6	52	103	32	0.339 ^4^
Yes	2	13	13	7	
Chewing betel nut (*n*)					
No	7	60	108	32	0.176 ^5^
Yes	1	5	8	7	
Obesity ^3^ (*n*)					
No	5	47	80	29	0.859 ^4^
Yes	3	18	36	10	

^1^ Married is defined as having a formal marriage relationship, and other refers to single, divorced, or widowed. ^2^ Drinking is defined as the regular consumption of alcoholic beverages more than once per week, not purely social drinking. ^3^ Obesity is defined as BMI ≥ 27. ^4^ Pearson’s chi-squared test. ^5^ Fisher’s exact test. * *p* < 0.05.

## Data Availability

The data of this study are not publicly available to protect the participants’ privacy.

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
