# Peer review of "Examining Health Literacy in Taiwanese Smoking Cessation Populations: A Multidimensional Evaluation of Sociodemographic Factors and Domain-Specific Competencies"

_healthcare, 2023, doi:10.3390/healthcare11162350_

Round 1

Reviewer 1 Report

This paper examines Health literacy in a population that attends a smoking cessation unit.Major concerns:

In principle, the subject is interesting since previous studies show that the personal resources of the person trying to quit smoking are a predictor of success in quitting tobacco use. However, the design of this study does not allow us to know the role played by Health literacy in quitting smoking. The authors only describe the statistical relationship between the different dimensions of Health literacy and certain demographic and social variables. But this does not provide new knowledge since, as the authors indicate in line 230, there are hardly any differences between the population studied and the general population. Likewise, the majority of the population studied had sufficient levels of Health literacy (lines 233-235).

It would have been interesting to study the influence of the different dimensions of Health literacy on smoking cessation in order to be able to design appropriate cessation programs for the different levels of Health literacy of the population wishing to quit smoking.

In the discussion chapter, much emphasis is placed on marriage since it is the only sociodemographic variable that is statistically significant in the population. The authors themselves point out that the associations between health literacy levels and background variables were generally weak (lines 267-8). However, even this statistical association is questionable. Isn't it possible that the age of the subjects is causing confusion? The married subjects are, on average, older than the unmarried subjects and in that time they may have obtained more personal resources, including Health literacy. It is a pity that the authors did not perform a stratified analysis of age to rule out this possible confounding.

Minor concerns:

The authors should have better reviewed their work before submission. Striking inconsistencies are detected in the text. For example, line 162 indicates that 33% of the population is married. However, Table 1 indicates that this percentage rises to 75.9%. Something similar occurs in Table 4 where the categories of the sex variable are inverted.

Lastly, the conclusions section should be eliminated or significantly reduced, since the sentences included therein reproduce texts from the discussion.

Reviewer 2 Report

·       Abstract:

o   The statement “No previous papers analyzed health literacy components in patients undergoing smoking cessation interventions (line 22-23)”

·       Introduction:

o   The introduction is too much, please be more concise, mainly highlight the background, purpose, and significance of the study, and make a simple literature review of existing similar studies.

o   Please review how relevant information is cited.

·       Materials and methods:

o   Indicate how many potential participants were invited, what proportion accepted to participate, and what differences existed between those who participated and those who did not.

o   Include recruitment procedures.

o   please include if there was an institutional review board for the study or if ethical principles were followed.

·       Statistical Analysis:

§  Chi-Square test present limitations when analyzing group data with 0 in values. Please evaluate other alternatives (i.e. Fisher’s exact test) to evaluate associations among groups with 0.

·       Discussion

o   Limitations

§  The authors should comment on using simple descriptive statistics as a limitation of the study.

§  Also, there should be included a paragraph discussing the limitations and strengths of the study.

Reviewer 3 Report

In general, the study is understandable in all its sections.

Aspects of improvement:

The design must be stated in the abstract and in the methodology.

The type of sample must be stated in the methodology.

In results, the authors do not have smoking cessation data at the time of data collection? All the participants were in cessation programs, what was the percentage of people in total abstinence? It is not clear

In limitations, the authors should comment on the very small number of women in the study population. In fact, no limitations of the study are clearly discussed.

Reviewer 4 Report

See attached. Thank you for the opportunity to review!

Round 2

Reviewer 1 Report

I agree with the authors when they say in their answer (2) to point 1: “Identifying the predictive factors of low health literacy within this group can aid in the design of cessation services that more closely meet their needs, especially the lowHL ones” To do this, we must know the factors related to Health literacy that help quit smoking by people who want to quit smoking and those who do not help these people. Unfortunately the design of this study does not allow this. As the authors indicate in their response (4): “We are currently working on studies that delve into the roles of health literacy in smoking cessation. Hopefully, we can have satisfactory findings. And hopefully based on those findings, we wish to be able to design appropriate cessation programs for the different levels of Health Literacy of the population wishing to quit smoking.”

Neither this conclusion can be sustained: "Therefore, tailored follow-up programs might improve health literacy and care outcomes, particularly for those without family support", because we do not know whether those with a higher level of health literacy and those who are married drop out tobacco to a greater extent than the other population groups.

Therefore, from the perspective of health promotion workers and policy makers, this study is of no interest.

Response point 2. It is rare that age, which is usually related to the educational level of the population, is not shown to be significant in this analysis. Have you analyzed the correlation between marriage status and age? Single subjects tend to be younger, while married subjects are older. Have the authors performed regression analysis categorizing age by quartiles or quintiles?
